# Secondary Lymphedema: Clinical Interdisciplinary Tricks to Overcome an Intriguing Disease

**DOI:** 10.3390/biology12050646

**Published:** 2023-04-24

**Authors:** Sylvain Mukenge, Daniela Negrini, Ottavio Alfieri

**Affiliations:** 1Department of Cardiothoracic-Vascular Surgery, Vita-Salute San Raffaele University, 20132 Milano, Italy; alfieri.ottavio@hsr.it; 2Department of Medicine and Surgery, University of Insubria, 21100 Varese, Italy; daniela.negrini@uninsubria.it

**Keywords:** secondary lymphedema, venous abnormality, long-term management, lymphatic microsurgery, lymphatic graft, lymphatic perioperative mesh

## Abstract

**Simple Summary:**

The lymphatic vascular system drains fluid, solutes and cells from the interstitial tissue surrounding the cells and subsequently returns the newly formed lymph back into blood circulation. Through their continuous drainage, lymphatic vessels maintain the adequate volume and composition of the tissue environment around the cells, guaranteeing their physiological behavior. Secondary lymphedema is a complex pathology whose progression depends upon an impairment of the lymphatic draining function, developing because of demolishing oncological surgeries, radiation, blood vascular malformations or surgical excision of local lymph nodes. In such cases, removal, obstruction, or impairment of the lymphatic vasculature cause the accumulation of fluid, solute and cell debris into the tissue, with progressive swelling, functional impairment and physical distress of the patient. The present review illustrates various cases of secondary lymphedema of increasing severity and the newest therapeutic solutions, spanning from indication of adequate body posture to physical therapy, to the newest and least invasive microsurgical approaches developed with long term satisfactorily results in the treatment of secondary lymphedema.

**Abstract:**

Secondary lymphedema is a complex pathology which is very impairing to the patient, consisting of fluid accumulation in the tissue, accompanied by alteration of the interstitial fibrous tissue matrix, deposition of cellular debris and local inflammation. It develops mostly in limbs and/or external genitals because of demolishing oncological surgery with excision of local lymph nodes, or it may depend upon inflammatory or infective diseases, trauma, or congenital vascular malformation. Its treatment foresees various approaches, from simple postural attitude to physical therapy, to minimally invasive lymphatic microsurgery. This review focuses on the different types of evolving peripheral lymphedema and describes potential solutions to single objective symptoms. Particular attention is paid to the newest lymphatic microsurgical approaches, such as lymphatic grafting and lympho-venous shunt application, to successfully heal, in the long term, serious cases of secondary lymphedema of limbs or external genitals. The presented data also emphasize the potential role of minimally invasive microsurgery in enhancing the development of newly formed lymphatic meshes, focusing on the need for further accurate research in the development of microsurgical approaches to the lymphatic vascular system.

## 1. The Lymphatic Vascular System and its Impairment

The vascular lymphatic system is specifically developed to guarantee the constancy of volume and solute composition of plasma and extracellular interstitial fluid. This target is achieved through a constant finely controlled removal of fluid, solutes of any size, and even cells from the interstitial tissue; the newly formed lymph is then carried along the lymphatic vessels to be eventually discharged into the venous blood. Lymph formation at tissue level requires that the hydraulic pressure be higher in the interstitium than in the lumen of initial lymphatics [1,2]; subsequent lymph propulsion along the lymphatic conduits is sustained by intraluminar pressure gradients developing between adjacent lymphatic segments, called lymphangions. The pressure regimes, which sustain lymph formation and propulsion, is set and maintained by both local tissue movements [3,4], such as heart activity and related cardiogenic oscillations [5], and respiratory muscle contraction [6], as well as by the action of synchronous contraction of muscle cells in the wall of collecting lymphatics. These same mechanisms may adapt [7,8] to increase lymph flow by up to 20–30-fold its physiological value, to prevent and/or counteract the accumulation of interstitial fluid whose volume may expand, for example, because of an augmented fluid filtration across the blood microvasculature wall [1,9]. However, an efficient control of tissue fluid volume is guaranteed only until attainment of a saturation threshold [1,10], beyond which fluid filtration rate across the wall of blood capillaries overwhelms the maximal draining capability of the local lymphatic network, leading to a progressive development of tissue edema. Lymphedema is a chronic complex impairment of tissue fluid homeostasis, depending not only upon an increased fluid filtration rate from blood capillaries, but, more significantly, upon the disorganization of the structure of the interstitial fibrous matrix and of initial and collecting lymphatic vessels with absent or greatly impaired lymphatic drainage. 

Lymphedema is classified, based on its clinical manifestations and morphology, into five evolutionary stages: -Stage 1—subclinical lymphedema, due to anatomical or functional impairment of lymphatic network detected through lymphoscintigraphy, but with no clinical evidence of edema-Stage 2—characterized by permanent edema, which worsens with orthostatic posture and partly improves with clinostatic position and/or rest-Stage 3—permanent edema, not improving with clinostatic posture, and progressively deteriorating with lymphangiotic-erisipeloid and/or peri-lymphangiosclerotic events of the interstitial tissue-Stage 4—fibrolymphedema with columnar limb, pachydermia and cutaneous verrucoses-Stage 5—elephantiasis with limb deformation, functional impairment, and alteration of skin trophysm (micosis, ulcers, lymphostatic verrucosis).

Depending upon the underlying cause, lymphedema can be further defined as primary or secondary [11]. Primary lymphedema often presents a complex and uncertain ethiology, attributable to (a) an evolutive chronic vascular pathology, (b) a congenital lymphangiodysplasia, (c) deleterious genetic mutations of lymphangiogenesis. The incidence rate of the disease in the population is about 1/10,000, with a male-to-female ratio of about 1/3. Based on its onset age, primary lymphedema can be classified as hereditary or idiopathic. The former may be present at birth (as in Milroy disease) or manifest itself from puberty on (as in Meige syndrome). Idiopathic primary lymphedema may also appear at various ages, from neonatal (congenital) to adult. 

In addition, primary lymphedema phenotypes may be multifocal, depending upon edema localization, its clinical manifestations, and mutations of the genes involved in the expression of the disease [11]. Based on patients’ evaluation [12], primary lymphedema has been classified as:(1)syndromic, when lymphedema is not the primary clinical evidence(2)with either pre-or postnatal systemic or visceral involvement(3)associated to growth, cutaneous or vascular abnormalities(4)congenital (i.e., Milroy’s disease)(5)with late onset (i.e., lymphedema distichiasis).

At variance with primary, secondary lymphedema does not depend upon genes’ mutations, but rather upon demolitive oncological surgery of bladder, uterus, breast, prostate or ovary, and of excision of pelvic, inguinal, or axillary lymph nodes. It may also accompany traumatic lesions and inflammatory or infectious diseases with involvement of major lymph nodal stations. Finally, impairment of otherwise healthy lymphatic collectors may be secondary to blood vessels’ malformation or pathways’ alterations. 

Considering the localization and severity degree, lymphedema may heavily interfere with the patients’ social life, with episodes of self-imposed isolation, psychological denial of the medical significance of their condition, and rejection of proper specific treatment or follow-up. In this respect, it is worth noting that there are only few medical units specifically oriented to the multidisciplinary treatment of lymphedema. Indeed, although lymphedema is usually not life-threatening, its clinical approach may be very cumbersome, as most of the lymphatic vascular network is clinically inaccessible, due to its complexity, tiny dimensions, fragile structure, and depth into surrounding tissue. At present, only relatively large and superficial lymphatic-collecting vessels can be ichnographically identified and may therefore be approached, either with appropriate physical therapy or with extremely specialized surgery. In particular: -in upper and lower limbs, the lymphatic network originates from initial lymphatics and, subsequently, merge into larger lymphatic collectors which carry the lymph to the axillary and inguinal lymph nodes, respectively. Through the thoracic duct, post-nodal lymph is eventually conveyed into the subclavian veins-lymphatics vessels draining the hypogastric area, the vaginal tunica, and the scrotal tissue discharge, respectively, into pre-aortic, para-aortic and superficial inguinal lymph nodes, respectively-lymphatic vessels originating from testicular tissue pass through the spermatic funiculus to reach the external iliac lymph nodes.

The following sections describe various types of chronic localized lymphedema of increasing severity and focus on the physical therapies and the newest frontiers of lymphatic microsurgeries available for successful, long-term treatment of patients. At present, clinical treatment of primary lymphedema almost exclusively consists of physical therapy. In fact, its multifocal expression and our still limited knowledge of its pathogenesis precludes the guarantee of any other indication. However, rare patients with congenital lymphedema, if over 35 years old, negative for genetical aberrations, and with only minor lymphatic vessels defects (degree Stage 1–3, documented through lymphoscintigraphy), may benefit from surgical procedures. 

Excluding the other already mentioned etiologies of peripheral lymphedema, we would like to start describing an extremely rare example of local lymphedema, due to a blood vascular morphological malformation and vessels’ pathway alteration.

## 2. Transitory Secondary Left Neck Latero-Cervical Lymphedema in a Rare Venous Plexus Malformation

Intrinsic artero-venous malformations and/or vessels pathway alterations due to embryonal angiogenetic mutations of distal vascular system usually do not lead to lymphatic insufficiencies, particularly when the vascular defect pertains to small vessels. However, lymphatic impairment may sometime develop and require clinical attention, such as in case of a transient secondary lymphedema of the neck, due to an extremely rare chronic retromandibular vein course, in a patient with known idiopathic double vena cava syndrome. The retromandibular vein is a major vein of the face, resulting from the confluence of the head and neck veins (temporo-maxillary vein and posterior facial vein). It descends in the substance of the parotid gland and, subsequently, divides into: (a) an anterior branch, which travels forward and joins the anterior facial vein to form the common facial vein, which, at last, drains into the internal jugular vein; (b) a posterior branch, which joins to the posterior auricular vein to form the external jugular vein. 

In this case, a 39-year-old woman in uneventful status came to our attention with transient left latero-cervical swelling. The main genes involved in primary lymphedema (including Milroy disease, Emberger syndrome, lymphedema-distichiasis, and other syndromes) were excluded. A vague relapsing edematous symptomatology appeared for the first time when she was 30 years old and occasionally emerged: (a) in clinostatic, either right or left, lateral decubitus, with the ipsilateral limb stretched over the head; (b) in orthostatic posture, with prolonged rotation to the right of the head-neck, right limb abduction, or inclination of the head to the left. Tissue swelling of left latero-cervical and supra-clavicular area, associated to patient discomfort, began after about one hour and spontaneously disappeared three hours after its development, without impairing the loco-regional or distal neuro-muscular function. Local physical anamnesis revealed no acute or chronic sequela of lymphedema. 

Previous echo-cardiography and color-doppler ultrasounds of supra-aortic trunks evidenced mild tissue edema in proximity of the neuro-vascular fasciculus, with: (a)double superior vena cava defect(b)no sign of increased right atrial pressure(c)minimum bicuspid valve prolapse, with uneventful hemodynamic consequences(d)significantly reduced diameter of the left versus the right external jugular vein(e)negative dynamic maneuverers to check for the thoracic outlet syndrome.

After careful and deep clinical evaluation to collect the most significant and reliable diagnostic information, we further investigated the patient through the total body computed axial tomography (CT), focusing on the thorax and, mainly, the mediastinum, with and without contrast medium. Photo Dynamic Eye (PDE) [13,14] was also performed to identify lymphatic collectors and follow their pathways. Briefly, PDE requires the injection, through an insulin needle, of ≈0.5 mL of indocyanine green into the subcutaneous tissue at the 2nd interdigital space of the hand, ipsilateral to the lymphedema location, followed by local massage. After a few minutes, with the aid of a near-infra-red camera, it is possible to clearly observe, on a monitor screen, the profile of the lymphatic vessels highlighted by the absorbed fluorescent dye. 

Tomographic imaging confirmed double superior cava vein disorder with normal empty of the right cava in the ipsilateral atrium, and anomalous outlet of the left cava in the enlarged coronary sinus. A 3-D reconstruction of CT sections (See Appendix A) clearly showed, immediately after the confluence of the left cava with the hemiazygos vein, a chronical ectasia of the left retromandibular vein that, rather than normally merging with the internal jugular vein, ran caudally to the left-medial side, anterior to the ipsilateral thyroid lobe. Then, it pointed to the right, running posterior to the medial epiphysis of the right clavicula, and emptied into the right subclavian vein (Figure 1). No evidence of adenopathy and/or soft tissue swelling was observed bilaterally in the supra-clavicular area.

The overall profile of lymphatic collectors, and of lymph node chains of the left neck, axilla and upper limb, were investigated through PDE lymphography [14,15]. Fluorescent indocyanine green, injected in the subcutaneous tissue of the left retro-auricular area (Figure 2), was very slowly drained through the main cervical lymphatic collector, covering the distance (∆l = 14 cm) to reach the supraclavicular lymph nodes (SCLN) in 180 s (∆t), with a flow velocity (vel=∆l/∆t) of ≈0.08 cm/s. In the ipsilateral normal upper limb, indocyanine green was injected at the second interdigital area and was propelled along the brachial lymphatic collector for 32 cm to reach the epitrochlear lymph nodes in 12 s, with a flow velocity of ≈2.6 cm/s. Since both the cervical and brachial collectors eventually discharge the lymph in the thoracic duct and then in the left subclavian vein, the observed difference in flow velocity suggests an impairment of lymph propulsion along the principal cervical collector. However, the latter was still clearly able to propel fluid to proximal lymph nodes (Figure 3).

The morphological abnormality does not seem, per se, to be able to explain the observed transitory edema in the left supraclavicular neck area. In addition, the lymphographic images of the left upper limb and of the axillary lymph nodes show a conserved lymphatic transit. The slowed transport of indocyanine green observed with PDE lymphography (Figure 2) indicated that the left cervical lymphatic flow was anyway maintained (Figure 3), ruling out the existence of permanent lymphatic suffering.

Although transitory, the observed swelling presents a documented evolution (Figure 2 and Figure 3) towards a chronic condition, whose pathogenesis is attributable to abnormal compression of the left cervical collector by the anomalous retromandibular vein (Figure 1). Indeed, such venous malformation [16] causes obstruction of the distal access into the contralateral subclavian vessel, with consequential venous hypertension and ectasia of the upstream retromandibular vein.

Lymphatic compression may determine, on one hand, an increase in lymphatic viscous flow resistances and, on the other, the enlargement of upstream lymphatic segments. This latter event would, in turn, cause: (1) a significant decrease of contractile strength of lymphatic smooth muscles and an impaired lymphatic propulsion [17], and/or (2) incontinency of the unidirectional valves between lymphangions. These abnormalities might all lead to lymphatic insufficiency, determining the development of distal loco-regional edema.

The sequence of event described above may be exacerbated by patient posture during:(a)acute extrinsic compression [1,3,6] at the uneasy access of the retromandibular into the right subclavian vein, upon a strong contraction during the articulation of the sternocleidomastoid-clavicular muscular branch,(b)clavicular rotation itself, at a given decubitus, and, last but not least,(c)intrinsic venous maneuvers of self-squirm, upon movements of upper limbs or neck lateral rotation.

In absence of severe disease symptoms, no therapeutic indication was provided, only suggesting specific postural adjustments of upper body segments. Edema worsening might eventually be treated with a multidisciplinary approach. At the four-year follow-up from our initial patient care, this advice helped the patient to successfully prevent the severity and discomfort of the occasional edematous events. In case of clinically worsening lymphedema, one might consider the advantages offered by the recently developed lymphatic vessels microsurgery [1,13,14]. The latter aims at assessing a new functional outlet of one or more lymphatic collectors, able to guarantee peripheral tissue drainage and long-term patency of the lymphatic route. At present, the commonly used super-microsurgical lympho-venous shunt remains the first and best indication.

## 3. Physical Therapy of Limb Lymphedema

The target of physical therapy is to improve lymphedematous tissue drainage by enhancing local lymph propulsion, and by recruiting shunted pre-existing lymphatic vessels. If lymphedema is localized in the upper or lower limbs, a possible approach consists of the so-called Complex Decongestive Therapy, including manual lympho-drainage, garments, physical exercise, skin care and prevention [18]. Therapy includes a first decongestive phase, whose target is to reduce the volume of the edematous limb, followed by a second phase, aiming at optimizing and maintaining of improvements attained in the decongestive phase. The latter is performed through:(a)manual lympho-drainage, consisting of superficial massages of edematous tissues to accelerate lymph transport towards the lymphatic network of healthy tissues. These maneuvers restore the drainage of the lymphedematous limb by recruiting collateral lymphatic vessels and/or shunts, able to vicariate the function of the armpit or groin lymph nodes resected, for example, during lymphadenectomy. In the specific case of post-mastectomy lymphedema, the attempt is to propel the lymph towards the terminus, the parasternal, paravertebral, contralateral axillary, ipsilateral inguinal and intercostal lymph nodes. Manual lympho-drainage presents few absolute contraindications, such as: (a) breasts’ neoplasm with systemic secondary localizations not surgically treated, or treated only with chemotherapy; (b) acute infections, inflammations and thrombosis, in which lympho-drainage could mobilize the limb thrombus, leading to pulmonary embolism or central nervous system stroke. Relative contraindications, in which lympho-drainage may be performed with adequate precautions, are, instead, pathologies such as systemic edema of cardiac or renal origin, or chronic infection, such as tuberculosis or toxoplasmosis;(b)compression garments, acting as a customed semirigid scaffold, exerting a progressively lower distal-to-proximal compression onto the lymphedematous limb (Figure 4). While the pressure exerted by the hosiery is low and well-tolerable at rest, muscular contraction associated with limb movement increases local tissue pressure, thus enhancing lymph propulsion [3,19] towards proximal lymph nodes. Being a real therapy, compression hosiery application also implies some contraindications, mostly related to sensitivity polyneuropathy, hemiplegia, etc.;(c)physical exercise, possibly performed from the first postoperative days, to prevent development of post-surgically edema, optimizing the effect of manual lymphodrainage, and of compressive bandage. Exercises include correct breathing, stretching, self-mobilization and muscular strengthening;(d)intermittent pneumatic compression, or pressotherapy, is performed through a cuff applied to the lymphedematous limb and pressurized according to specifically costumed clinical protocols under close supervision of specialized health personnel. It represents an efficient therapy when accompanied by lymphodrainage and compressive hosiery, but it cannot replace them. Pressotherapy presents the same contraindications as manual lymphodrainage.

**Figure 4 biology-12-00646-f004:**
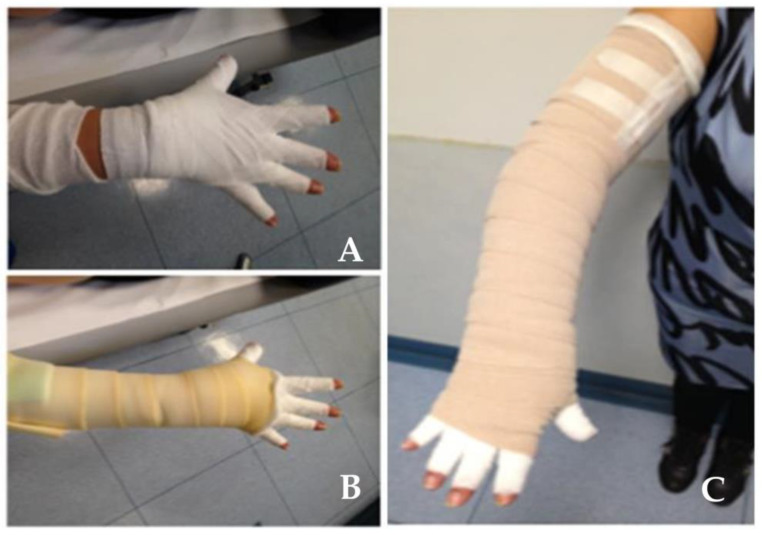
Physical therapy of lymphedema. Subsequent steps in confectioning compressive garments of the hand (**A**), arm (**B**) and complete upper limb (**C**).

If limb volume is satisfactorily decreased and stable, one can move to the maintenance phase, implying:(a)use of a compressive-elastic brace, a crucial element of the therapy, with a function similar to compressive garments;(b)accurate skin care, to prevent ulcerations and frequent complications, such as development of erysipelas, fever, itching, skin erythema, limb heat and pain. In addition, it is of absolute importance to wear the compressive garments prescribed at the hospital discharge.

In case physical therapy does not satisfactorily reduce lymphedema and/or the degree of patient discomfort, one may consider the opportunity of looking to a specific surgical approach. The latter exploits various procedures aimed at recanalizing the lymphatic ducts (lympho-lymphatic shunt), whose pathway has been obstructed, and/or at realizing a low resistance inlet of a lymphatic vessel directly into a peripheral vein (lympho-venous shunt).

## 4. Surgical Approach to Peripheral Lymphedema: From Microsurgical Experimental Trials to Clinical Applications

Microsurgery may be described as a multidisciplinary branch of surgery, derived from experiences developed on laboratory animals to approach tiny organic structures with dimensions in the order of microns; it is performed through the aid of an intraoperative microscope, microsurgical instruments, and suture stitches from 9–0 to 11–0.

In the last 50 years of the past century, the development of manufactory industries was accompanied by a parallel increase of the occurrence of serious blunt trauma in workplaces, with increasing episodes of limb and/or finger amputations. In addition, the increased use of motorbikes, especially among young people, sometimes driven at exceedingly high velocities, determined the increase of serious road accidents, resulting in severe lesions of the nervous system (e.g., to the brachial plexus) and/or of the bone-muscle-joints complexes, with involvement of both large vascular trunks and microvasculature of limbs or extremities. To face the increased need of new strategies for the treatment of these complex clinical conditions, either in emergency or as a specific focused service, new dedicated Trauma Centers developed, with the aim of offering an adequate and modern multidisciplinary approach to enable the patients to recover their total functional integrity and be fully reinstated into society. In parallel, experimental studies on laboratory animals involved an increasing number of surgical areas: from general surgery to orthopedics, neurosurgery, vascular and thoracic surgery, gynecology, and from experimental organ transplants to ophthalmological surgery. In this context, the first microsurgical artero-arterial (femoro-femoral), veno-venous (cavo-caval) and nervous (initial sutures of superficial nerves, such as the sural nerve) anastomosis were perfected in rats. In the following years, to develop new approaches able to avoid palliative demolitive interventions in the treatment of peripheral lymphedema, surgeons started focusing their attention on the fine anatomy, the physiology, and the physiopathology of the vascular lymphatic system. New techniques were perfected in laboratories to set up all types (termino-lateral or termino-terminal, venous grafts, etc.) of lympho-venous anastomosis [20]. In addition, other insights, such as lympho-lymphatic shunts, lymphatic “ileo-mesenteric-inguinal bridges” [21], lymphatic grafts [22] and lymph nodes grafts with their own vasculature [23], were developed in qualified international research centers. Meanwhile, bio-molecular and anatomo-pathophysiological research focused on several basic aspects of the lymphatic system, including the development of germinal lymphatic cells, the mechanisms of lymph formation and propulsion, the description of lymphatic cell markers and receptors, andthe identification of the genetic steps controlling the lymphatic system’s development and function. In surgeons’ minds, these studies were of great interest, because they might have hopefully provided the scientific background to make it possible to overcome the uncertainties inherent in the classical microsurgical approach of secondary lymphedema. That is, the dilemma of guaranteeing healing of vessels undergoing microsurgical operations; indeed, even using the most accurate techniques and the most sophisticated instrumentation, confection of anastomosis exposes the vessel wall layers to considerable biological upheaval. Accurate experimental morphological studies [24] performed through scanning electron microscope (Figure 5) verified, indeed, the efficiency of the microsurgical procedures, even in the long term, thus guaranteeing its safe applicability and consequent biological healing in patients.

Preoperative screening for microsurgery of lymphedema includes, in addition to the feedback of the physical examination and imaging diagnostics, the data collection of medical history, including (a) remote, recent and family objective cardiovascular and neurological pathology, (b) allergies, habits and diet, and (c) current medical therapies, in particular, on the use of Ca^++^-antagonists.

Microsurgery of the upper or lower limbs requires lymphoscintigraphy and a color doppler ultrasound, to rule out the concomitant occurrence of other vascular diseases, such as, for example, deep vein thrombosis. Instead, lymphedema of external male genital organs can be instrumentally diagnosed only through echography. The following sections illustrate the current microsurgical approaches and their long-term outcomes.

### 4.1. Lymphatic Grafting

This procedure is usually performed to solve cases of secondary lymphedema of the upper limbs, but it may also be applied to lower limbs. First, one must proceed with the identification of the major lymphatic collector of the healthy lower limb. With the patient deeply anesthetized, 1 mL of Patent Blue dye is injected in the 2nd interdigital space of the same limb. Under stereomicroscopic view, the lymphatic collector is identified and excised (Figure 6), while preserving up to four branching vessels.

After setting up and identifying the lymphatic-lymph node structures of the supraclavicular-ipsilateral area, and of the upper third of the medial arm in the proximal axillary area, the lymphatic graft is then inserted though axillary subcutaneous tunneling (Figure 6), allowing for the anastomosis sutures [22]. During the graft maneuver, at first, lymph flows out of the distal stump; when graft insertion is successfully completed, the outflow of lymph from the graft stops and fluid flows along the inserted segment. The optimum is to prepare at least 4 grafts proximally and, likewise, distally, sutured with 10–0 or 9–0 Vicryl thread, which is rapidly absorbed. The complete surgery may last 4 to 6 h. Antibiotics (7 days) and subcutaneous low molecular-weight heparin (30 days) is required post-surgically. Patients who, alternatively, undergo arm lympho-venous anastomosis need an equally high number of anastomosis (4 to 6) to guarantee the efficiency of limb drainage. The postoperative pharmacological therapy is the same.

### 4.2. Lympho-Venous Shunt in Lower Limb

The surgical approach to identify the lymphatic collectors and venous vessels of upper and lower limbs and genitalia is identical. In lower limbs, super-microsurgery protocol [25] allows for identification of the subcutaneous lymphatic vessels through minimally invasive multiple skin incisions along the limb, facilitating the preparation of multiple anastomoses, as in lymphatic grafting. Previously, an incision is made at the inguinocrural area to allow identification of the saphenous vein, or its branching, and the lymphatic collectors (Figure 7). Under stereomicroscopic view (magnification, ×40), and using 10–0 or 9–0 Vicryl stitches, up to four anastomoses can be confectioned between prenodal lymphatics of the crural area and collaterals (termino-terminal), or with the saphenous vein itself (termino-lateral), respectively. Low-molecular-weight heparin must be used during surgery to reduce the risks of thrombotic occlusion of the lymph vessel or of the vein at the anastomosis site. The postoperative pharmacological protocol is the same as for previous techniques.

### 4.3. Bilateral Lympho-Venous Shunt of the Spermatic Cord

Under general anesthesia, following inguinoscrotal, cremaster and vaginal tunica incisions, the funicular lymph vessels (diameter ≈500 mm) and cord veins can be observed bilaterally under stereomicroscopic view. These collectors are used to perform either lymphovenous termino-lateral or termino-terminal anastomosis, depending upon the proximity and pathway of the pampiniform veins, which discharge in the ipsilateral spermatic vein, and of the spermatic lymphatic collectors. After removal of vein adventitia [14], following common procedure in confection of venous anastomosis, up to four anastomoses to each side can be performed, depending on the number of lymphatic collectors found, considering that several anastomoses improve the chances of effective drainage (Figure 8). The postoperative pharmacological protocol is the same as for previous techniques.

### 4.4. Combined Swelling of Male External Genitals and Lower Limbs

In the lower limb, the lymphatic system is organized in: (a) a superficial network, merging into lymphatic collectors directed to the inguinal lymph nodes, from which the lymph is carried through the iliac collectors into the cisterna chili and, thereafter, to the thoracic duct and, eventually, into the subclavian veins; (b) a deeper network, whose collectors follow the sub-fascial limb neuro-vascular plexus to reach the peri-aortic lymph nodes and the thoracic duct.

Lymphedema of male external genitals is often accompanied by unilateral or bilateral chronic swelling of the lower limbs. As such, the microsurgical approach consists of a two-stage treatment: first, genital microsurgery, and, after six months, if needed, lympho-venous shunt of the lower limb (Figure 9).

The results of this approach clearly indicate that the shunt of the spermatic cord in the treatment of genital edema was often accompanied by a significant improvement of the edema of the lower limbs, such as to lead us, after a few cases, to subsequently exclude the two-step surgery. Improvement of limb lymphedema after shunt of the spermatic cord may depend upon the fact that, after excision of inguinal lymph nodes, lymph from lower limbs is shunted through the scrotal and testicular pathways. Similarly, when testicular drainage is restored, edema of the scrotum and penis is also significantly drained, possibly through direct connections between scrotal lymphatics and testicular networks.

### 4.5. Long Term Outcome of the Microsurgery of Upper and Lower Limb, and of Spermatic Cord

A positive long-term postsurgical recovery of the patient critically depends upon the maintenance of anastomosis patency. The degree of recovery at 180 days from surgery in subjects treated for arm or lower limb lymphedema may be examined using: (a) noninvasive PDE lymphography (upper limb, Figure 10A–G; lower limb, Figure 11A–H), (b) ability to normally use the limb, (c) lack of pain or erysipelas, (d) return to normal lifestyle, and (e) determination of limb volume [15]. The latter can be calculated from direct measurement of the perimeter of both patient’s limbs at fixed standard points, on the assumption that the limb has the shape of a truncated cone [15,26]. 

Patients can be considered “responders” if, at 6 months after surgery, limb volume decreased by >60% of the preoperative lymphedematous volume.

In a previous study from our Lymphedema Center [15], in patients treated for upper limb lymphedema, the presurgical edematous limb volume was about 40% greater compared to the contralateral normal limb (Figure 10I).

At 180 days from surgery, absolute limb volume decreased by 35% in responding patients, with excess limb volume decreasing by about 120% (Figure 10L). A slight, not significant, limb volume reduction was instead observed in nonresponding patients [15].

In those with lower limb lymphedema, the swelling increased by ≈40%, with respect to the other normal limb (Figure 11L). At 180 days, in responding patients, the edematous limb volume was reduced to a value even slightly smaller than that of the control contralateral limb (Figure 11M). Instead, in nonresponding patients, lymphedema only slightly, but not significantly, improved at 6 months after surgery (Figure 11I–M).

About 70% of patients undergoing microsurgery of the spermatic cord attained a 95% 100% scrotal swelling remission at 6 months after surgery (Figure 12). In all responding patients, patent subcutaneous lymphatic ducts could be clearly identified (Figure 12C–E). With respect to the pre-operative condition, tissue fluid volume, as well as hypogastric abdominal wall thickness, were significantly reduced, palpability of the testis was normal with decreased weight sensation, with decreased diuresis and disappearance of episodic erysipelas. Recovery of the preoperative scrotal size may sometimes require additional plastic surgery to remove the excessive scrotal skin.

A set of patients only partially improved after the procedure, with ≈60% assessment at 180 days from surgery.

### 4.6. Perianastomotic Lymphatic Meshes

In addition to maintenance of patent lymphatic collectors, a common observation in responding patients is the development of complex lymphatic meshes, well documented by PDE (Figure 11F), and/or by magnetic resonance (Figure 12) in the region surrounding the anastomosis. These nets are detectable, starting from about three months from surgery, and seem to further develop at later times.

Lymphatic meshes did not apparently develop in nonresponding patients, nor in the area around lymphatic grafting when the anastomosis was confectioned with two biologically similar tissues, such as terminal stumps of lymphatic collectors. This clinically relevant phenomenon requires further investigation to clarify the biologic processes of acute lymphangiogenesis after tissue trauma or surgery. Indeed, at present, it is not clear whether these new vessels reflect passive recruitment of previously not utilized vessels or new active sprouting.

### 4.7. Long-term Lymphatic Vessel Development

In several tissues, lymphatic vessels may merge into networks shaped like closed loops in the diaphragm [22,23], or arcades in the intestine [2]. Loops and arcades are morphological tissue-specific structures genetically encoded in the physiological lymphatic development to accommodate the tissue need and anatomical conformation: indeed, they might act as reservoir between initial and collecting vessels. The perianastomotic lymphatic networks documented in Figure 13 are, instead, not encountered physiologically, but develop as microsurgery-induced local lymphatic sprouting to increase the draining capacity of the vascular lymphatic network. This type of observation is not new in lymphedema; indeed, in their study on therapeutic regeneration of mouse lymphatic and immune cell functions, Lenti et al. showed that, upon transplantation of lympho-organoid at the site of resected lymph nodes, lympho-organoid integrated into the endogenous lymphatic vasculature and efficiently restored lymphatic drainage and perfusion [27]. Lymphatic cell biology has received much attention in recent years [28,29]; however, at present, the mechanisms controlling lymphangiogenesis in pathophysiological condition are still not completely unveiled. Studies [30,31] on the effect of tissue trauma on local lymphatic networks have shown that, in rats’ tails, the maintenance of interstitial fluid flux upstream to a tissue lesion may promote lymphangiogenesis and reconstruction of a putative lymphatic pathway guaranteeing proper distal tissue drainage. In addition, studies on primary lymphedema models [32] suggest that lymphatic function may be strictly correlated to the collagen and lipid content of the tissue matrix.

## 5. Conclusions

Lymphedema is a complex pathology of the lymphatic system that, regardless of its origin, may present different clinical manifestations. Physical therapy must be tried, at first, in the treatment of both primary and secondary lymphedema; only after its failure, other procedures must be considered. Particularly, in cases of secondary lymphedema, the bio-technological advances and/or refinement of more and more accurate surgical instruments, as well as the development of new surgical procedures, allowed for the attainment of satisfactorily results, in line with the required clinical criteria. Indeed, with respect to the previously widely utilized demolitive oncological interventions, the microsurgical approach minimized the relative sequelae or iatrogeneses. Interestingly, in patients successfully treated with microsurgery of spermatic and lower limb lymphatics, new lymphatic meshes develop to optimize perianastomotic tissue drainage. Hence, at variance with more destructive surgery, minimally invasive microsurgery might promote local neo-lymphangiogenesis. Therefore, more research studies are needed to further refine the bioengineering development associated to microsurgical approaches that not only reduce the patient hospitalization and recovery period, but seem to facilitate, with respect to more invasive surgical procedures, the lymphatic and extracellular matrix fibrous tissue repair.

## Figures and Tables

**Figure 1 biology-12-00646-f001:**
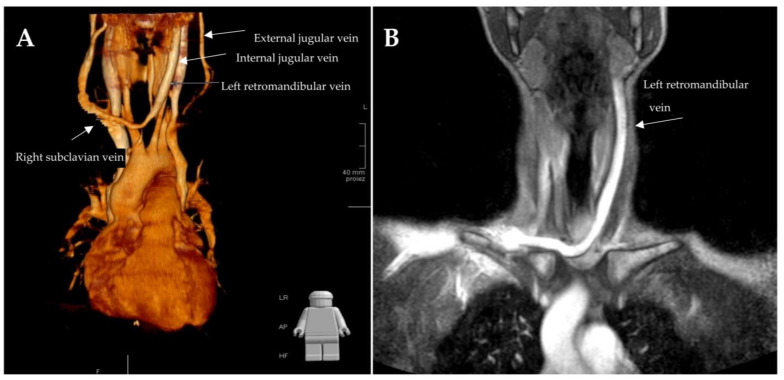
(**A**) Three-dimensional chest and neck tomography reconstructions confirm the double vena cava malformation. An additional anomaly affects the retromandibular left vein that appears ectasic, with an abnormal pathway emptying into the right subclavian vein. (**B**) Maximum intensity projection of an Angio Magnetic Resonance sequence in venous phase after Gadolinium injection: the left retromandibular vein, instead of normally emptying into the ipsilateral internal jugular vein, crosses the midline under the superficial facial plane and over the pre-thyroidal muscles (not depicted in the angiogram) and merge to the confluence of the right internal jugular and subclavian veins behind the right sternoclavicular joint.

**Figure 2 biology-12-00646-f002:**
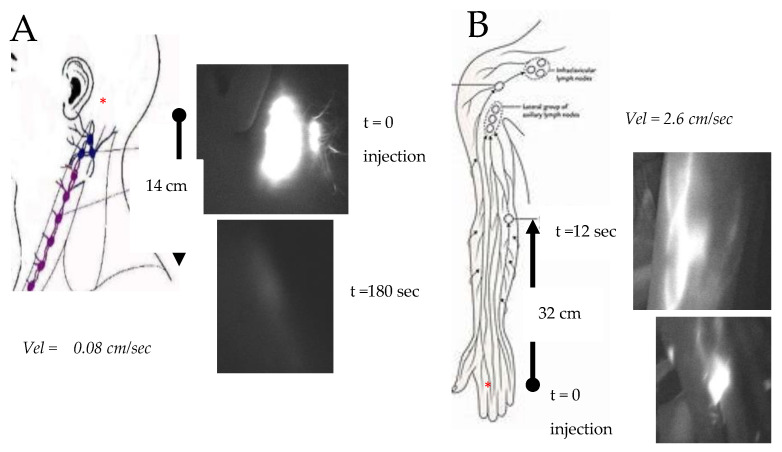
PDE lymphography showing the distribution of a bolus of fluorescent indocyanine green injected (red asterisks) into the subcutaneous tissue of the left retro-auricular area (**A**) or in the left second interdigital area (**B**), respectively. The dye travelled at a flow velocity of 0.08 cm/s along the main cervical collector directed to the supraclavicular lymph node (SCLN, white asterisk) and of 2.6 cm/s in the lymphatic collectors’ satellites of the basilic vein to the epitrochlear lymph nodes, as indicated by black arrows.

**Figure 3 biology-12-00646-f003:**
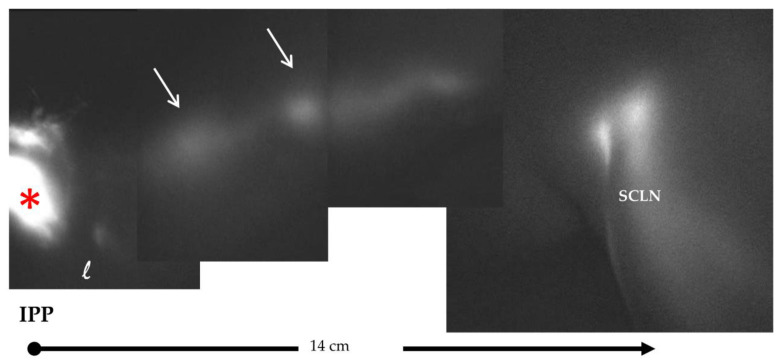
Horizontal reconstruction of indocyanine green lymphography (PDE) image showing, at 1 h from injection, the spreading of the fluorescent dye from the injection point (IP and red asterisk) at the retro-auricular subcutaneous tissue along the left latero-cervical lymphatic collector to reach the supraclavicular lymph nodes (SCLN). Although the injected dye eventually reaches the SCLN, its spreading is extremely delayed and the lymphatic collector appears irregular in shape, with abnormal enlargements (arrows) along its pathway.

**Figure 5 biology-12-00646-f005:**
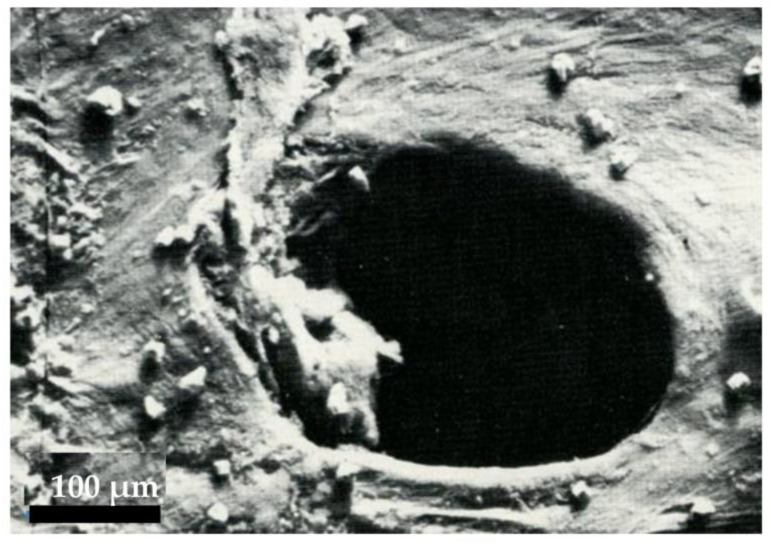
Electron scanning microscopy image documenting the patency of a lympho-venous anastomosis after 3 months from microsurgery in experimental rat.

**Figure 6 biology-12-00646-f006:**
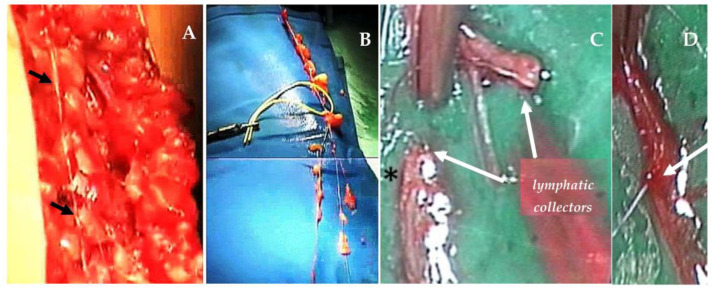
Sample of major lymphatic collector of the thigh ((**A**), black arrows) and subsequent graft preparation (**B**). Confection of a termino-terminal lympho-lymphatic anastomosis between a lymphatic collector of the superior third of the arm ((**C**), asterisk) and the distal branches of the graft, deriving from the supraclavicular lymph nodes, or from collectors themselves. (**D**), definitive anastomosis (white arrow).

**Figure 7 biology-12-00646-f007:**
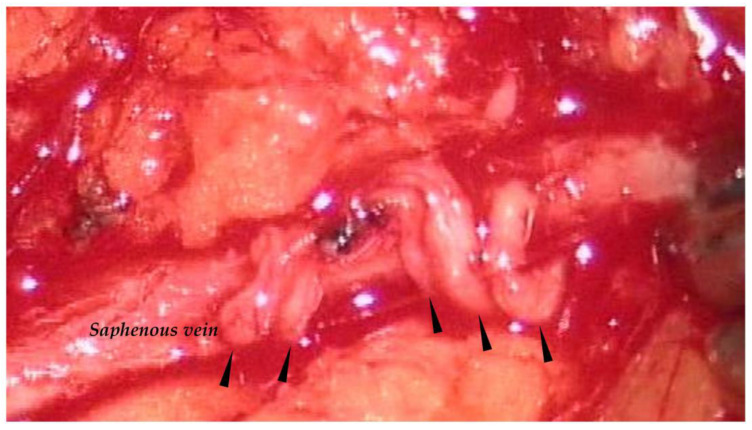
Lympho-venous shunt of the lower limb. Arrows point to anastomizedlymphatic collectors emptying into the saphenous vein.

**Figure 8 biology-12-00646-f008:**
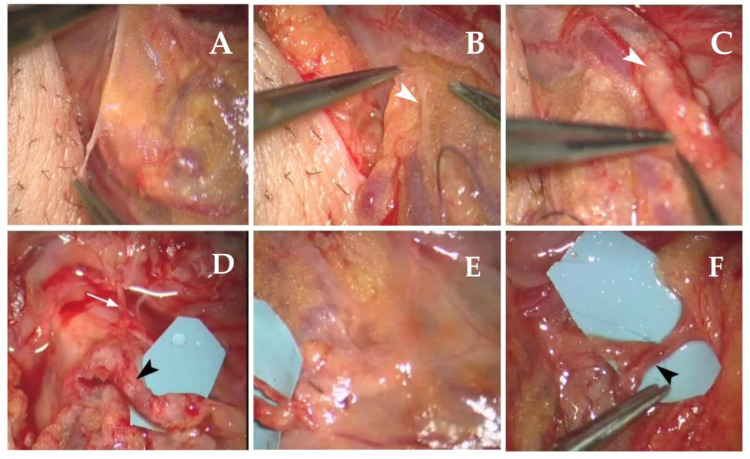
(**A**) Incision of the scrotal skin with evidence of spermatic funiculum; (**B**) lymphatic duct (at white arrow); (**C**) adventitiectomy of the pampiniform vein in setting up for anastomosis (white arrow); (**D**) shunt between a lymphatic duct (white arrow) and the pampiniform vein, shown in panel C (black arrow); (**E**) other testicle, lymphatic duct and vein prepared for anastomosis; (**F**) final anastomosis (black arrow). Obtained shunting from the lymphatic trunk and the pampiniform vein, shown in panel E. Suture thread and needle are visible passing through the anastomosis.

**Figure 9 biology-12-00646-f009:**
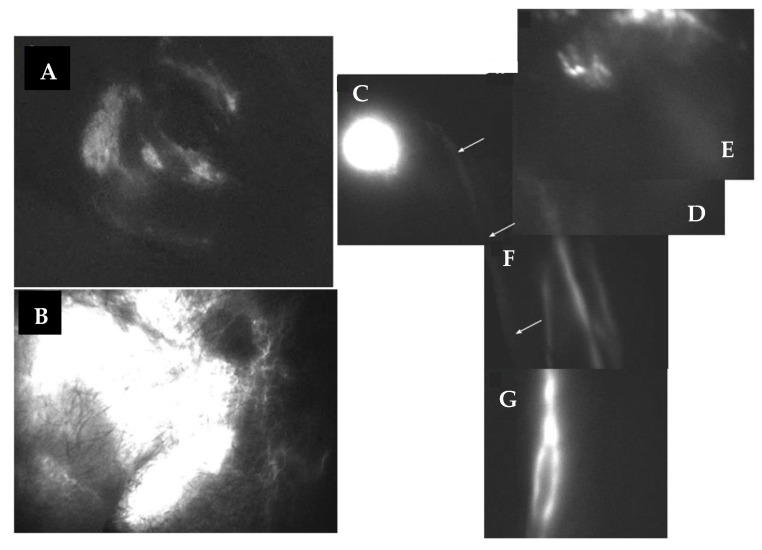
Photodynamic eye (PDE) lymphography shows that, in normal testis (**A**), the injected dye is canalized into the surrounding lymphatic network and is rapidly drained out of the subcutaneous scrotal tissue. Vice versa, in lymphedematous testis (**B**), the dye diffusively accumulates in the scrotal subcutaneous tissue because of the impaired lymphatic drainage. Panels C to G show a PDE sequential reconstruction of the pathway, followed by indocyanine injected in the testis (**C**), and, 1 h after, in the foot of a patient previously suffering of lymphedema of both external genitals and lower limb, after three months after undergoing a lympho-venous shunt of the spermatic cord. Images show that: (1) after injection in the testis, indocyanine is rapidly drained away from the injection site; (2) after injection in the foot, indocyanine flows along lymph vessels from the lower limb (**D**,**E**) to be shunted directly into the surgically reconstructed lymphatic route, emptying directly into the spermatic vein (**F**,**G**), bypassing the occlusion of the excised pelvic-inguinal lymph-node chain. White arrows help identifying the path of a deep scarcely visible collector.

**Figure 10 biology-12-00646-f010:**
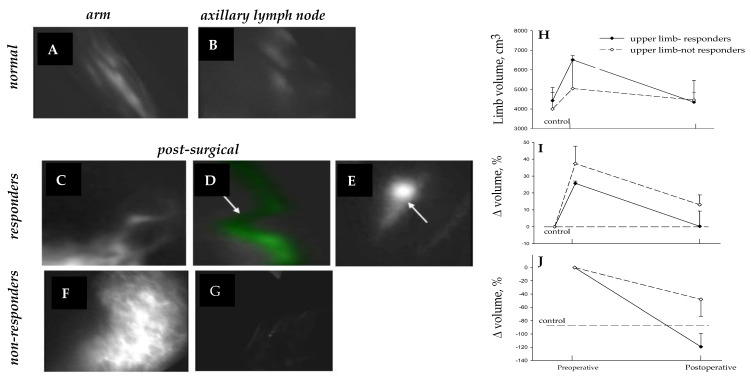
Lymph collectors at 180 days from lymphatic microsurgery in subject affected by arm lymphedema. PDE images document normal (**A**,**B**), responding (**C**–**E**), and nonresponding (**F**,**G**) patients. Limb volume (**H**), change in volume with respect to contralateral normal limb (**I**) and percentage volume recovery with respect to preoperative edematous limb volume (**J**) show that in responding subjects, lymphedema was essentially removed, while it persisted in non-responders. White arrows identify a patent lymphatic vessel (**D**) and an axillary lymph node reached by the dye in a responding patient.

**Figure 11 biology-12-00646-f011:**
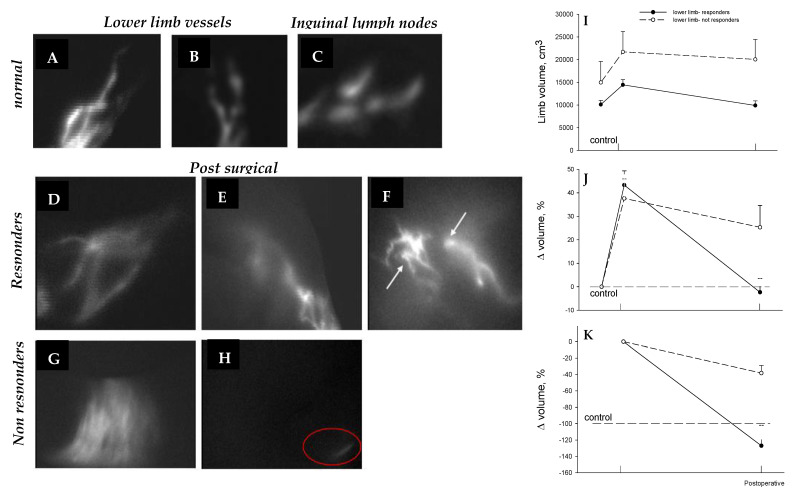
PDE images of lower limb at 180 days from surgery in normal (**A**–**C**), responding (**D**–**F**), and nonresponding (**G**,**H**) patients. Lymphatic meshes developed (arrows) between lymphatic collectors, and around the lympho-venous anastomosis of responding, but not of not-responding, patients (red oval in panel **H**). Limb volume (**I**), change in volume with respect to contralateral normal limb (**J**) and percentage volume recovery with respect to preoperative edematous limb volume (**K**) show that that in responding patients lymphedema was completely solved, while non-responders only partially improved after lymphatic surgery.

**Figure 12 biology-12-00646-f012:**
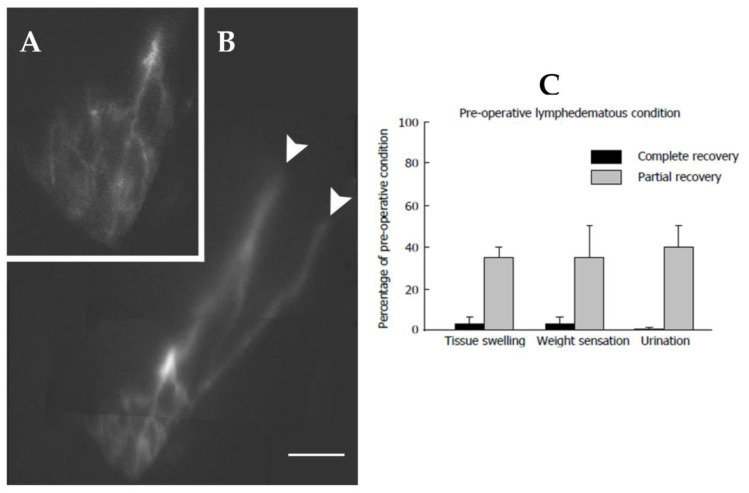
(**A**) PDE image showing the lymphatic network of the scrotum at 6 months after a lymphovenous derivation for the treatment of scrotal lymphedema. At the sites of lympho-venous anastomosis (white arrows), the dye fluorescence disappears because: (1) the canalized lymphatic collectors run deeper into the tissue to reach the pampiniform veins, and (2) downstream of the patent lympho-venous anastomosis, the dye is discharged into the spermatic vein, thus, being no longer visible. (**B**) magnified and digitally enhanced still photo of the lymphatic network of the testicle shown in panel (**A**). (**C**) Post-surgical self-evaluation of tissue swelling, weight sensation and urination expressed as percentage of pre-operative lymphedematous condition in patients showing a complete or only partial recovery at three months after surgery. Based on patients’ evaluation, even partial recovery represented a significant improvement, with respect to pre-operative condition. In responding patients, the objective clinical outcome was accompanied by a subjective evaluation of almost complete recovery of the normal tissue swelling, weight sensation and urination. Zero percentage on the Y axis corresponds to the normal condition. Histograms present mean values ± SD of the mean.

**Figure 13 biology-12-00646-f013:**
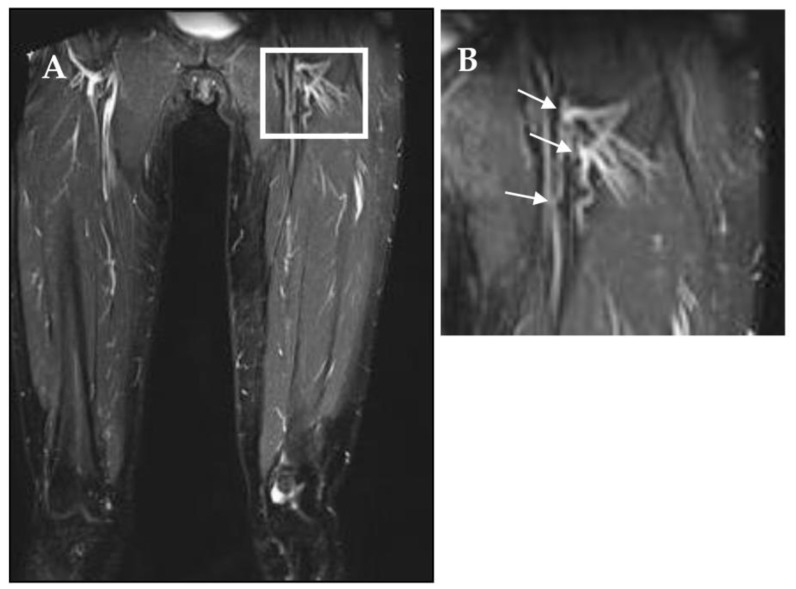
(**A**) Magnetic resonance image of the lower limb obtained at 8 years after microsurgery in a patient who underwent to lymphatic microsurgery to confection a latero-terminal lympho-venous anastomosis between lymphatic vessels, and the saphenous vein in proximity of the saphenous femoral vein cross. (**B**) Details of the left limb crural area (white rectangle): white arrows identify the sites where latero-terminal lympho-venous anastomosis were confectioned. On comparing the left vs. the right crural areas, an evident mesh of either recruited or sprouting lymphatic vessels in the peri-anastomotic area was clearly observable.

## Data Availability

Not applicable.

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
