# Peer review of "Secondary Lymphedema: Clinical Interdisciplinary Tricks to Overcome an Intriguing Disease"

_biology, 2023, doi:10.3390/biology12050646_

Round 1

Reviewer 1 Report

Dear Authors

I enjoyed very much your quite thorough comprehensive review on the current status of chronic lymphedema, wisely(?) limited only to the secondary lymphedema, management. But I feel short on one specific issue you cannot simply stay away, wishing you to address more clearly on its relationship with the primary lymphedema at the beginning through Introduction with a few additional paragraphs once and for all.

Indeed, currently, primary lymphedema is wrapped up together with secondary lymphedema to undergo summary execution as one group of chronic lymphedemas, but they are as different as apple and orange, if not more, with entirely different prospects for future management strategy.

Because, primary lymphedema is mostly of congenital vascular malformation origin limited to the lymphatic system as one of lymphatic malformations so that its future management strategy also should accommodate its prospect as one of lymphatic malformations, and your current review on its management limited to the secondary lymphedema should be modified in this regard to providing more clear bird’s eye view on chronic lymphedema as a whole.

1. Line 89: recommend that ‘fibrous-fatty tissue’ alone would bring unnecessary confusion with the risk of misunderstanding. It should be reinforced with the addition of ‘regional lymph nodes’ to connect to the next sentence.

2. Line 91-92: Current statement ‘impairment of lymphatic collectors may depend upon vascular malformation or pathways alterations’ for the primary lymphedema deserved to get further clarification through this Introduction section as suggested through the general overview as above.

3. Line 229: “the popliteal lymph nodes, as indicated by black arrows”???? Verify it properly.

4. Line 238: Ditto! “the popliteal lymph nodes”???

5. Line 242-243: I have some reservation to accept/agree with your conclusion of “ --- the low flow velocity and of the presence of irregular ecstatic sites along its pathway” without control group data, preferably the contralateral/right side to compare. Since you do NOT have such data to confirm your conclusion, you should include proper narration to verify this liability.

All the best,

A Reviewer

Author Response

Answers to Reviewer # 1

First of all, we wish to thank Reviewer #1 for pointing at the lack of information on primary lymphedema in the previous version of the text and suggesting including more information. We have now inserted new paragraphs (lines 82-99) in Section 1 and throughout the text to address this point. 

Please note that, as suggested by Reviewer #2, the title has been slightly modified.

We prefer to use the American spelling and therefore in the revised version we modified all terms from “oedema” into “edema” and “lymphoedema” in “lymphedema” throughout the text.

  1. Line 89: recommend that ‘fibrous-fatty tissue’ alone would bring unnecessary confusion with the risk of misunderstanding. It should be reinforced with the addition of ‘regional lymph nodes’ to connect to the next sentence.

The whole paragraph has been rewritten, “fibrous – fatty tissue” has been substituted by “lymph nodes”

  1. Line 91-92: Current statement ‘impairment of lymphatic collectors may depend upon vascular malformation or pathways alterations’ for the primary lymphedema deserved to get further clarification through this Introduction section as suggested through the general overview as above.

The text has been modified to clarify the significance of our statement, including clearer reference to primary lymphedema.

  1. Line 229: “the popliteal lymph nodes, as indicated by black arrows”???? Verify it properly.
  2. Line 238: Ditto! “the popliteal lymph nodes”???

Thank you for noticing these very obvious mistakes that we have now corrected

  1. Line 242-243: I have some reservation to accept/agree with your conclusion of “ --- the low flow velocity and of the presence of irregular ecstatic sites along its pathway” without control group data, preferably the contralateral/right side to compare. Since you do NOT have such data to confirm your conclusion, you should include proper narration to verify this liability 

We rephrased the text to cope with the Reviewer criticisms, deleting the sentence “ the low flow velocity and of the presence of irregular ectasic sites along its pathway” .

Reviewer 2 Report

Dear authors,

Congratulations on writing such a comprehensive review. I have a few suggestions for you to take into consideration.

- The title feels misleading compared to what the actual review is about. Please reconsider to specify that there is a lot of surgical information in the text. 

- Please reconsider using parts to reference so you do not loose the focus in the paper by describing something in detail which has been published and accepted widely (for example the paragraphs on truncated cone formula).

- Sometimes as a reader I completely lose sight of what you as authors are trying to explain. It is written very elaborately and therefore I can not understand what the focus is 

- Please be consistent is use of of words that match other work published in lymphoedema such as compression garments, compression hosiery, IPC devices and decide on the use of American or British spelling such as the use of 'lymphoedema' or American spelling 'lymphedema'.

- Please make sure you reference appropriately when making claims. 

Best of luck

Author Response

We thank Reviewer #2 for the constructive suggestions and criticisms. We have revised the text accordingly, trying to make the text simpler and clearer. In particular, please note that, as suggested by Reviewer # 1, we have included a paragraph in Section 1 to add more information on primary lymphedema.

- The title feels misleading compared to what the actual review is about. Please reconsider to specify that there is a lot of surgical information in the text.

As suggested, the title has been slightly modified

- Please reconsider using parts to reference so you do not loose the focus in the paper by describing something in detail which has been published and accepted widely (for example the paragraphs on truncated cone formula).

We have shortened same paragraphs and deleted the section with the equations of limb volume calculation, leaving the adequate references , as suggested.

- Sometimes as a reader I completely lose sight of what you as authors are trying to explain. It is written very elaborately and therefore I can not understand what the focus is

We tried to make the text simpler and clearer throughout the text

- Please be consistent is use of of words that match other work published in lymphoedema such as compression garments, compression hosiery, IPC devices and decide on the use of American or British spelling such as the use of 'lymphoedema' or American spelling 'lymphedema'.

Thank you for pointing to these details.  We prefer to use the American spelling and therefore modifies all terms from “oedema” to “edema” throughout the text.

- Please make sure you reference appropriately when making claims.

Done.

Round 2

Reviewer 2 Report

Thank you for making the necessary adjustments